

# Impact of enzymatic digestion on bacterial community composition in CF airway samples

Kayla M. Williamson[1], Brandie D. Wagner[1,2], Charles E. Robertson[3], Emily J. Johnson[2,4], Edith T. Zemanick[2] and J. Kirk Harris[2]

[1] Department of Biostatistics and Informatics, Colorado School of Public Health, University of Colorado, Aurora, CO, United States of America
[2] Department of Pediatrics, School of Medicine, University of Colorado, Aurora, CO, United States of America
[3] Division of Infectious Diseases, School of Medicine, University of Colorado, Aurora, CO, United States of America
[4] Multicare Tacoma Family Medicine, Tacoma, WA, United States of America

Corresponding author
Brandie D. Wagner,
brandie.wagner@ucdenver.edu

## ABSTRACT

**Background**. Previous studies have demonstrated the importance of DNA extraction methods for molecular detection of *Staphylococcus*, an important bacterial group in cystic fibrosis (CF). We sought to evaluate the effect of enzymatic digestion (EnzD) prior to DNA extraction on bacterial communities identified in sputum and oropharyngeal swab (OP) samples from patients with CF.

**Methods**. DNA from 81 samples (39 sputum and 42 OP) collected from 63 patients with CF was extracted in duplicate with and without EnzD. Bacterial communities were determined by rRNA gene sequencing, and measures of alpha and beta diversity were calculated. Principal Coordinate Analysis (PCoA) was used to assess differences at the community level and Wilcoxon Signed Rank tests were used to compare relative abundance (RA) of individual genera for paired samples with and without EnzD.

**Results**. Shannon Diversity Index (alpha-diversity) decreased in sputum and OP samples with the use of EnzD. Larger shifts in community composition were observed for OP samples (beta-diversity, measured by Morisita-Horn), whereas less change in communities was observed for sputum samples. The use of EnzD with OP swabs resulted in significant increase in RA for the genera *Gemella* ($p < 0.01$), *Streptococcus* ($p < 0.01$), and *Rothia* ($p < 0.01$). *Staphylococcus* ($p < 0.01$) was the only genus with a significant increase in RA from sputum, whereas the following genera decreased in RA with EnzD: *Veillonella* ($p < 0.01$), *Granulicatella* ($p < 0.01$), *Prevotella* ($p < 0.01$), and *Gemella* ($p = 0.02$). In OP samples, higher RA of Gram-positive taxa was associated with larger changes in microbial community composition.

**Discussion**. We show that the application of EnzD to CF airway samples, particularly OP swabs, results in differences in microbial communities detected by sequencing. Use of EnzD can result in large changes in bacterial community composition, and is particularly useful for detection of *Staphylococcus* in CF OP samples. The enhanced identification of *Staphylococcus aureus* is a strong indication to utilize EnzD in studies that use OP swabs to monitor CF airway communities.

## INTRODUCTION

Cystic Fibrosis (CF) is an autosomal recessive disease, characterized by chronic airway infection, whose predominant pathogens include *Staphylococcus aureus, Pseudomonas aeruginosa* and other Gram-negative bacteria (*Heijerman, 2005*; *LiPuma, 2010*). OP swabs are commonly used when children are unable to expectorate (*Zemanick et al., 2015*). This is particularly true for pediatric, non-expectorating subjects with CF in whom oropharyngeal (OP) cultures are used as a surrogate for lower airway bacteria (*Zemanick et al., 2015*). In the pediatric studies, OP swabs have ranged from 30% to 68% of samples collected (*Armstrong et al., 1996*; *Zemanick et al., 2010*; *Wolter et al., 2013*; *Hoppe et al., 2015*) and therefore represent an important sample type in early CF.

Sequencing is becoming more widely used to evaluate bacterial communities in CF airway samples (*Cummings et al., 2016*). DNA preparation prior to sequencing, including cell lysis, can have a profound effect on microbial community composition especially in detecting certain organisms such as *S. aureus* (*Zhao et al., 2012*; *Yuan et al., 2012*; *Willner et al., 2012*; *Lozupone et al., 2013*; *Pérez-Losada et al., 2016*). S. *aureus* in particular has a rigid cell wall that can be hard to rupture, hindering the ability to efficiently extract DNA for molecular detection (*Zhao et al., 2012*).

Previous studies have indicated that use of enzymatic digestion (EnzD) may enhance the ability to detect *Staphylococcus* (*Schindler & Schuhardt, 1964*; *Browder et al., 1965*; *Yuan et al., 2012*; *Zhao et al., 2012*; *Johnson et al., 2016*). Specifically, *Yuan et al. (2012)* performed a comprehensive experiment evaluating multiple DNA extraction methods utilizing human associated bacterial species as well as a mock community. *Zhao et al. (2012)* found that EnzD increased the yield of *Staphylococcus* in sputum samples. Similarly, *Johnson et al. (2016)* determined EnzD improved the sensitivity of sequencing in OP swabs while no differences were observed in sputum samples. Increased sensitivity with EnzD was robust when compared to clinical culture results and qPCR. Furthermore, the work showed that the majority of *Staphylococcus* was *S. aureus*. It remains unclear, however, how the use of EnzD effects the remaining bacterial community in clinical CF samples. In this work, we build on these two previous CF studies to evaluate the effect of DNA extraction using enzymatic digestion (EnzD) on bacterial communities detected from oropharyngeal swab (OP) and sputum samples collected from patients with CF.

## MATERIALS & METHODS

### Patient demographics and samples

Sputum and OP samples were obtained from patients with CF as part of standard of care for monitoring bacterial infection during routine patient visits. Further explanation of selection of samples to be used for this study are included in a previous publication (*Johnson et al., 2016*). Excess specimen was stored frozen at −80 °C for molecular assessment of infection. Standard sputum processing protocol was performed for sample homogenization utilizing sputalysin and standard CF culture was performed following CF Foundation guidelines (*Burns et al., 1998*). The Colorado Multiple Institutional Review Board approved the study

(COMIRB 07-0835). Written informed consent was obtained from patients or guardians. Written informed assent was obtained for children 10–17 years of age.

## DNA extraction methods

DNA was extracted using the Qiagen EZ1 Advanced automated extraction platform using the Tissue kit and bacterial card per manufacturer's instructions. EnzD was performed on one replicate of each sample by mixing with lysostaphin (final concentration 0.18 mg/mL) and lysozyme (3.6 mg/mL) and incubated at 37 °C for 30 min. Samples were then digested with proteinase K (1.4 mg/mL) and incubated at 65 °C for ten minutes, then incubated at 95 °C for 10 min (*Zhao et al., 2012*; *Johnson et al., 2016*). Lysozyme and lysostaphin target degradation of the bacterial cell wall by targeting peptidoglycan (lysozyme) and pentaglycine bridges (lysostaphin) (*Salazar & Asenjo, 2007*). Lysostaphin is specific to the subset of staphylococci that contain pentaglycine bridges including the human pathogen *S. aureus* (*Trayer & Buckley, 1970*). Proteinase K is a broad specificity endopeptidase that is used to digest proteins (*Gradisar et al., 2005*).

## 16S rRNA gene amplicon library construction

Bacterial profiles were determined by broad-range amplification and sequence analysis of 16S rRNA genes following our previously described methods (*Hara et al., 2012*; *Markle et al., 2013*). Amplicons were generated using primers that target approximately 300 base pairs of the V1/V2 variable region of the 16S rRNA gene. Each DNA was amplified in triplicate along with a barcode specific negative PCR control. PCR reactions contained 1X HotMaster Mix (5Prime), 150 nM each PCR primer and template in a reaction volume of 25 µl. Cycling conditions were 94 °C denaturation for 120 s followed by 30 cycles of 95 °C 20 s, 52 °C 20 s and 65 °C 60 s. Amplification was confirmed using agarose gel electrophoresis. None of the negative PCR controls showed evidence of amplification. PCR products were normalized based on agarose gel densitometry, pooled, lyophilized, purified and concentrated using a DNA Clean and Concentrator Kit (Zymo, Irvine, CA, USA). Pooled amplicons were quantified using Qubit Fluorometer *2.0* (Invitrogen, Carlsbad, CA, USA). The pool was diluted to 4 nM and denatured with 0.2 N NaOH at room temperature. The denatured DNA was diluted to 15 pM and spiked with 10% of the Illumina PhiX control DNA prior to loading the sequencer. Illumina paired-end sequencing was performed on the MiSeq using a 500 cycle version 2 reagent kit.

## Analysis of illumina paired-end reads

As previously described, paired-end sequences were sorted by sample via barcodes in the paired reads with a python script (*Markle et al., 2013*). Sorted paired end sequence data were deposited in the NCBI Short Read Archive under accession number SRP043334. The sorted paired reads were assembled using phrap (*Ewing & Green, 1998*; *Ewing, 1998*). Pairs that did not assemble were discarded. Assembled sequence ends were trimmed over a moving window of five nucleotides until average quality met or exceeded 20. Trimmed sequences with more than one ambiguity or shorter than 250 nt were discarded. Potential chimeras identified with Uchime (usearch6.0.203_i86linux32) (*Edgar et al., 2011*) using the Schloss Silva reference sequences (*Schloss & Westcott, 2011*) were removed from subsequent

analyses. Assembled sequences were aligned and classified with SINA (1.3.0-r23838) (*Pruesse, Peplies & Glöckner, 2012*) using the 479,726 bacterial sequences in Silva 115NR (*Quast et al., 2013*) as reference configured to yield the Silva taxonomy. Sequences with identical taxonomic assignments were grouped into Operational taxonomic units (OTUs). This process generated 4,302,223 sequences for 162 samples (average sequence length: 316 nt; average sample size: 26,557 sequences/sample; minimum sample size: 7,527; maximum sample size: 63,105). The median Good's coverage score was ≥99.70% at the rarefaction point of 7,527. The software package Explicet (v2.10.5, http://www.explicet.org) (*Robertson et al., 2013*) was used to calculate rarefied values for diversity measurements.

## Statistical analysis

Shannon-H alpha diversity, evenness, richness and Morisita Horn (MH) beta diversity were calculated in Explicet. MH beta diversity is a measure of similarity between two communities and ranges from 0 (no similarity) to 1 (identical communities). Principal Coordinate Analysis (PCoA) was used to assess differences at the community level utilizing 1-MH. 1-MH distances were used to show the dissimilarity between with EnzD and without EnzD in PCoA plots instead of MH, which are a measure of similarity. Relative abundance (RA) for each genus was calculated by dividing the genera-specific sequence counts by the total number of sequences obtained for each sample. Shannon-H diversity, evenness and richness were compared using a Wilcoxon Signed Rank test. Differences in phyla and genera between EnzD samples versus non-EnzD samples with a median RA of at least 1% were evaluated using Wilcoxon Signed rank test for paired samples. Benjamini–Hochberg corrections were used to account for multiple comparisons (*Benjamini & Yekutieli, 2001*). Spearman correlations were used to assess associations between MH for paired samples and RA of specific genera from the non-digested sample. Analyses were calculated using R version 3.2.4 Revised (2016-03-16 r70336).

## RESULTS

### Sample collection

We analyzed 81 airway samples (39 sputum and 42 OP, Fig. S1) collected from 63 patients with cystic fibrosis ranging in age from 1.5 to 24 years (9–24 sputum and 1.5–23 OP). Half of the subjects were female (51%), the majority of subjects were non-Hispanic white (89%) and 52% of subjects were homozygous F508. The median number of samples collected per subject was one and ranged between one and three samples. Seven samples (9%) were negative for standard CF pathogens by culture (Fig. S2).

### Changes in individual organisms

Individual taxa were assessed at the phyla and genera level by sample type (OP and Sputum). For OP samples, we observed a significantly lower RA for the phyla *Bacteroidetes, Fusobacteria* and *Proteobacteria* when EnzD was used. *Firmicutes* and *Actinobacteria* both were detected in higher RA with the use of EnzD. For sputum, the RA for *Bacteroidetes* and *Proteobacteria* were lower with EnzD (Table 1) consistent with findings in OP samples (Fig. 1A).

**Table 1  Median relative abundance in EnzD and Non-EnzD groups; estimated change (EnzD–Non-EnzD) in median RA by Phyla.**

| Phyla | EnzD% | Non-EnzD% | Estimated change in RA% | *P*-value | Corrected *p*-value |
|---|---|---|---|---|---|
| OP | | | | | |
| *Actinobacteria* | 5.60 | 1.25 | 3.79 | <0.01 | <0.01 |
| *Bacteroidetes* | 7.80 | 19.35 | −12.73 | <0.01 | <0.01 |
| *Firmicutes* | 68.31 | 48.32 | 16.78 | <0.01 | <0.01 |
| *Fusobacteria* | 3.50 | 8.83 | −1.97 | <0.01 | <0.01 |
| *Proteobacteria* | 2.88 | 7.71 | −5.52 | <0.01 | <0.01 |
| Sputum | | | | | |
| *Actinobacteria* | 0.03 | 0.01 | 1.12 | 0.06 | 0.06 |
| *Bacteroidetes* | 0.01 | 0.06 | −3.52 | <0.01 | <0.01 |
| *Firmicutes* | 0.84 | 0.72 | 4.77 | 0.08 | 0.08 |
| *Proteobacteria* | 0.01 | 0.02 | −1.36 | <0.01 | <0.01 |

Genera with a statistically significantly higher RA following EnzD for OP swabs consisted of: *Gemella*, *Streptococcus*, *Actinomyces*, *Johnsonella,* and *Rothia.* Genera with a statistically significantly lower RA for OP swabs consisted of: *Prevotella*, *Veillonella*, *Neisseria*, *Haemophilus*, *Leptotrichia*, *Campylobacter*, *Fusobacterium,* and *Granulicatella*. For these genera, while the changes are statistically significant, the changes may not be clinically meaningful. The largest changes in RA (>5%) are seen in *Streptococcus*, *Prevotella*, and *Veillonella* (Table 2). *Staphylococcus* was the only genus with a significantly higher RA in sputum following EnzD. Genera with a statistically significantly lower RA in sputum consisted of: *Veillonella*, *Granulicatella*, *Prevotella*, and *Gemella* (Table 2). The largest estimated change in RA for sputum samples is attributed to *Staphylococcus* (Table 2). *Gemella* was higher in OP samples with EnzD while it was lower in sputum samples with EnzD (Fig. 1B).

Genera traditionally associated with CF include *Pseudomonas, Staphylococcus, Haemophilus, Stenotrophomonas*, and *Achromobacter* (Zemanick et al., 2015). All of the samples have low RA for *Achromobacter* and all of the OP samples had low RA for *Pseudomonas* and *Stenotrophomonas*. In the sputum samples, 11 had low amounts of all CF pathogens, 11 were dominated by *Staphylococcus*, all of which increased with EnzD, and three were dominated mostly by *Haemophilus*. Of the remaining 14 sputum samples, three samples were dominated by *Stenotrophomonas*, one increased, one decreased and one remained unchanged with EnzD. Eleven sputum samples had >5% RA *Pseudomonas* prior to EnzD, the RA increased after EnzD in six of these samples (median 18.7 range 1.1%–68.4%), in those where RA decreased, the median and range was −6.1%, −1.2% to 42.0% (Table 3).

## Overall community changes

The Shannon-H alpha Diversity Index was significantly lower ($p < 0.01$) with EnzD in sputum samples (Fig. S3) due to higher RA of *Staphylococcus* resulting in a significantly less even community (Shannon-H, Wilcoxon Signed Rank, $p < 0.01$). For OP samples, the

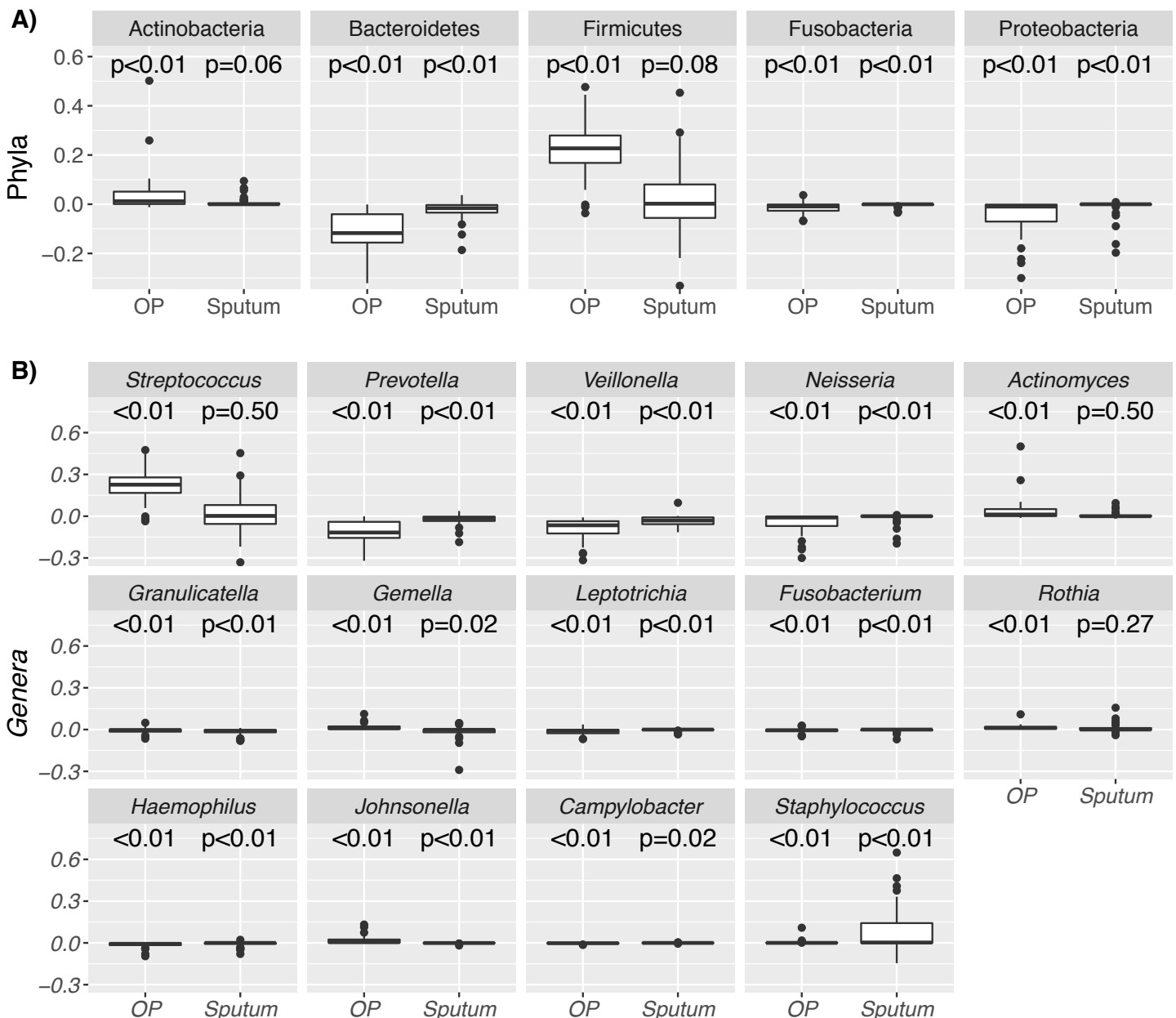

**Figure 1** **Boxplots for individual taxa that change within paired samples.** The distribution of differences in paired samples is shown for phyla (A) and genera (B). Taxa with limited differences between the paired samples are tightly distributed around zero, and those with increased relative abundance after EnzD have positive distributions. Note the median RA for Fusobacteria in sputum is less than 1%. *P*-values correspond to Benjamini–Hochberg corrected Wilcoxon Signed rank test for paired samples.

Shannon-H alpha Diversity Index was also lower likely due to higher RA of *Streptococcus* with EnzD, although this result did not reach statistical significance (Shannon-H, Wilcoxon Signed Rank, $p = 0.06$). Richness and evenness were also not significantly affected by EnzD in OP samples. Changes in alpha diversity due to use of EnzD do not necessarily correspond to changes in microbial community composition as measured using beta diversity (Fig. S4).

**Table 2  Median relative abundance in EnzD and Non-EnzD groups; estimated change (EnzD- Non-EnzD); in Median RA by Genera.**

| Genera | EnzD% | Non-EnzD% | Estimated change in RA% | P-value | Corrected p-value |
|---|---|---|---|---|---|
| OP | | | | | |
| *Streptococcus* | 36.51 | 14.81 | 21.75 | <0.01 | <0.01 |
| *Prevotella* | 6.62 | 16.20 | −10.90 | <0.01 | <0.01 |
| *Veillonella* | 9.27 | 19.22 | −7.86 | <0.01 | <0.01 |
| *Neisseria* | 0.72 | 1.94 | −3.62 | <0.01 | <0.01 |
| *Actinomyces* | 2.93 | 0.73 | 2.50 | <0.01 | <0.01 |
| *Granulicatella* | 1.80 | 2.44 | −0.62 | <0.01 | <0.01 |
| *Gemella* | 1.89 | 0.64 | 1.23 | <0.01 | <0.01 |
| *Leptotrichia* | 1.16 | 3.79 | −1.44 | <0.01 | <0.01 |
| *Fusobacterium* | 1.47 | 3.08 | −0.56 | <0.01 | <0.01 |
| *Rothia* | 1.70 | 0.27 | 1.25 | <0.01 | <0.01 |
| *Haemophilus* | 0.45 | 1.69 | −0.91 | <0.01 | <0.01 |
| *Johnsonella* | 0.97 | 0.40 | 1.19 | <0.01 | <0.01 |
| *Campylobacter* | 0.50 | 0.75 | −0.25 | <0.01 | <0.01 |
| Sputum | | | | | |
| *Streptococcus* | 27.85 | 22.48 | 1.33 | 0.50 | 0.50 |
| *Staphylococcus* | 1.39 | 0.62 | 5.47 | <0.01 | <0.01 |
| *Prevotella* | 0.88 | 3.54 | −1.91 | <0.01 | <0.01 |
| *Veillonella* | 1.09 | 6.01 | −3.55 | <0.01 | <0.01 |
| *Granulicatella* | 0.53 | 1.77 | −1.30 | <0.01 | <0.01 |
| *Gemella* | 0.53 | 1.68 | −0.62 | <0.01 | 0.02 |
| *Rothia* | 0.84 | 0.57 | 0.21 | 0.24 | 0.27 |

## Beta diversity changes

There was less impact of EnzD on community composition in sputum compared to OP samples as measured by MH beta diversity values for each sample pair. The majority of the MH values (79.5%) for sputum were greater than 0.8, whereas only 38.1% of OP samples were above that level. For the OP swabs the lowest MH was 0.4 for sputum samples the lowest MH was 0.48. The distributions between the two sample types were statistically different (MH, Wilcoxon Rank Sum, $p < 0.01$, Fig. 2). We further investigated these changes using the MH distance matrix in Principal Coordinate Analysis (PCoA). OP samples changed more consistently in ordination space with EnzD (Fig. 3A). The consistent changes observed in the OP samples were largely due to the change in *Streptococcus*, *Gemella*, and *Rothia* (all Gram-positive) as well as *Neisseria*, *Leptotrichia*, *Prevotella*, and *Veillonella* (Gram-negative); with the samples undergoing EnzD primarily clustered around *Streptococcus, Gemella*, and *Rothia* (Fig. 3A). Findings here align with previously discussed individual level changes in genera where Gram-positive bacteria were found in significantly higher RA while Gram-negative bacteria was found in significantly lower RA with EnzD (Table 2). As indicated in Fig. 3B, sputum samples showed less impact from EnzD, with a large group of samples having limited changes in MH. A subset of the sputum samples

**Table 3** **Displays the RA for CF pathogens detected in each sample.** Samples are highlighted when the difference in relative abundance was at least 1%, red for increase with EnzD and blue with a decrease. The grey rows indicate the median (min, max) for the difference in relative abundance for the highlighted samples.

| | Staphylococcus | | Achromobacter | | Haemophilus | | Stenotrophomonas | | Pseudomonas | |
|---|---|---|---|---|---|---|---|---|---|---|
| | Non EnzD | EnzD | Non EnzD | EnzD | Non EnzD | EnzD | Non EnzD | EnzD | Non EnzD | EnzD |
| OP | | | | | | | | | | |
| OP1 | 0 | 0.01% | 0 | 0 | 1.63% | 0.34% | 0 | 0 | 0 | 0 |
| OP2 | 0 | 0.08% | 0 | 0 | 4.03% | 1.56% | 0 | 0 | 0 | 0 |
| OP3 | 0.01% | 0.07% | 0 | 0 | 0.04% | 0 | 0 | 0 | 0 | 0 |
| OP4 | 0.01% | 0.03% | 0 | 0 | 1.96% | 0.56% | 0 | 0 | 0 | 0 |
| OP5 | 0.01% | 0.04% | 0 | 0 | 2.65% | 0.83% | 0 | 0 | 0 | 0 |
| OP6 | 0 | 0.01% | 0 | 0 | 0.00% | 0.01% | 0 | 0.01% | 0 | 0 |
| OP7 | 0 | 0 | 0 | 0 | 5.71% | 1.46% | 0 | 0 | 0 | 0 |
| OP8 | 0.15% | 1.13% | 0 | 0 | 2.05% | 0.65% | 0 | 0 | 0 | 0 |
| OP9 | 0 | 0.02% | 0 | 0 | 15.12% | 7.11% | 0 | 0 | 0 | 0 |
| OP10 | 0 | 0.04% | 0 | 0 | 0.25% | 0.08% | 0 | 0 | 0 | 0 |
| OP11 | 0.01% | 0.03% | 0 | 0 | 5.86% | 2.03% | 0 | 0 | 0 | 0 |
| OP12 | 0.02% | 0.03% | 0 | 0 | 0.21% | 0.11% | 0 | 0 | 0.01% | 0.02% |
| OP13 | 0.08% | 0.30% | 0 | 0 | 1.19% | 0.22% | 0.01% | 0 | 0 | 0 |
| OP14 | 0.06% | 0.37% | 0 | 0 | 2.41% | 1.43% | 0 | 0 | 0.03% | 0 |
| OP15 | 0.07% | 1.02% | 0 | 0 | 0.03% | 0.04% | 0 | 0 | 0.01% | 0.01% |
| OP16 | 0.04% | 0.05% | 0 | 0 | 0.09% | 0.02% | 0 | 0 | 0 | 0 |
| OP17 | 0.01% | 0.08% | 0 | 0 | 0.57% | 0.31% | 0 | 0 | 0 | 0 |
| OP18 | 0.01% | 0.01% | 0 | 0 | 3.49% | 1.87% | 0 | 0 | 0.01% | 0.01% |
| OP19 | 0.01% | 0.57% | 0 | 0 | 1.91% | 0.57% | 0 | 0 | 0 | 0.01% |
| OP20 | 0 | 0 | 0 | 0 | 3.45% | 1.79% | 0 | 0 | 0 | 0 |
| OP21 | 0.02% | 0.10% | 0 | 0 | 0.40% | 0.18% | 0 | 0 | 0.01% | 0.02% |
| OP22 | 0 | 0 | 0 | 0 | 0.35% | 0.05% | 0 | 0 | 0.01% | 0.05% |
| OP23 | 0.07% | 0.18% | 0 | 0 | 11.95% | 7.84% | 0.01% | 0 | 0 | 0.01% |
| OP24 | 0 | 0 | 0 | 0 | 0.17% | 0.03% | 0 | 0 | 0 | 0 |
| OP25 | 0.01% | 0.01% | 0 | 0 | 0.23% | 0.07% | 0.01% | 0.01% | 0.02% | 0.01% |
| OP26 | 2.17% | 3.92% | 0 | 0 | 0.45% | 0.04% | 0 | 0 | 0.01% | 0 |
| OP27 | 0.11% | 0.16% | 0 | 0 | 0 | 0 | 0 | 0 | 0 | 0.01% |
| OP28 | 0.05% | 0 | 0 | 0 | 2.44% | 1.82% | 0 | 0 | 0 | 0 |
| OP29 | 0 | 0 | 0 | 0 | 1.75% | 0.67% | 0.05% | 0.03% | 0 | 0 |
| OP30 | 0.01% | 0.03% | 0 | 0 | 40.38% | 30.81% | 0 | 0 | 0 | 0 |
| OP31 | 0 | 0 | 0 | 0 | 0.61% | 0.14% | 0 | 0 | 0 | 0 |
| OP32 | 0.01% | 0.01% | 0 | 0 | 3.40% | 1.09% | 0 | 0 | 0.07% | 0.06% |
| OP33 | 0 | 0 | 0 | 0 | 0.30% | 0.04% | 0 | 0 | 0 | 0 |
| OP34 | 3.34% | 14.25% | 0 | 0 | 2.01% | 1.19% | 0 | 0 | 0 | 0 |
| OP35 | 0.05% | 0.08% | 0 | 0 | 4.44% | 4.27% | 0 | 0 | 0 | 0 |
| OP36 | 0.02% | 0.22% | 0 | 0 | 0.03% | 0.01% | 0 | 0 | 0 | 0 |
| OP37 | 0.11% | 0.19% | 0 | 0 | 2.06% | 0.58% | 0 | 0 | 0 | 0 |
| OP38 | 0.01% | 0.04% | 0 | 0 | 3.68% | 1.54% | 0 | 0 | 0 | 0 |

**Table 3** (*continued*)

| | Staphylococcus | | Achromobacter | | Haemophilus | | Stenotrophomonas | | Pseudomonas | |
|---|---|---|---|---|---|---|---|---|---|---|
| | Non EnzD | EnzD | Non EnzD | EnzD | Non EnzD | EnzD | Non EnzD | EnzD | Non EnzD | EnzD |
| OP39 | 0.06% | 0.07% | 0 | 0 | 0 | 0 | 0 | 0 | 0.01% | 0 |
| OP40 | 0.13% | 0.11% | 0 | 0 | 0.01% | 0 | 0 | 0 | 0.01% | 0.02% |
| OP41 | 0.05% | 0.03% | 0 | 0 | 1.91% | 1.20% | 0 | 0 | 0 | 0 |
| OP42 | 0.01% | 0.01% | 0 | 0 | 0.05% | 0.01% | 0 | 0 | 0.02% | 0.01% |
| | 6.3 (1.8, 10.9) | | − | | −1.8 (−1.1, −9.6) | | − | | − | |
| Sputum | | | | | | | | | | |
| ES1 | 2.24% | 1.99% | 0 | 0 | 0 | 0 | 0 | 0 | 0.66% | 0.51% |
| ES2 | 0.01% | 0.05% | 0 | 0 | 2.89% | 2.79% | 0 | 0 | 39.12% | 79.91% |
| ES3 | 0 | 0 | 0 | 0 | 8.77% | 0.83% | 0 | 0 | 0 | 0.01% |
| ES4 | 0.05% | 0.12% | 0 | 0 | 9.52% | 6.47% | 0 | 0 | 1.27% | 1.02% |
| ES5 | 34.61% | 72.13% | 0 | 0 | 0 | 0 | 0 | 0 | 0 | 0.01% |
| ES6 | 43.91% | 84.67% | 0 | 0 | 0.01% | 0 | 0 | 0 | 5.04% | 1.39% |
| ES7 | 89.38% | 97.47% | 0 | 0 | 0 | 0 | 0.07% | 0.02% | 0.01% | 0 |
| ES8 | 16.72% | 32.21% | 0 | 0 | 0 | 0 | 0 | 0 | 52.01% | 9.96% |
| ES9 | 0.49% | 8.24% | 0 | 0 | 1.53% | 0.71% | 0 | 0 | 26.46% | 33.92% |
| ES10 | 0.20% | 0.35% | 0 | 0 | 0.54% | 0.09% | 0 | 0 | 0 | 0.01% |
| ES11 | 31.47% | 96.31% | 0 | 0 | 4.11% | 0.23% | 0.14% | 0.02% | 0.01% | 0 |
| ES12 | 0.04% | 0.04% | 0 | 0 | 0 | 0 | 0 | 0 | 10.08% | 3.89% |
| ES13 | 26.05% | 11.45% | 0 | 0 | 1.08% | 0.72% | 0 | 0 | 31.81% | 58.28% |
| ES14 | 0.06% | 0.13% | 0 | 0 | 0 | 0 | 0 | 0 | 0.02% | 0.03% |
| ES15 | 16.08% | 29.44% | 0 | 0 | 11.39% | 7.50% | 0 | 0 | 13.04% | 14.15% |
| ES16 | 1.48% | 7.26% | 0 | 0 | 2.26% | 1.25% | 0 | 0 | 52.82% | 63.76% |
| ES17 | 67.38% | 86.70% | 0 | 0 | 0.17% | 0.02% | 0 | 0 | 1.69% | 0.88% |
| ES18 | 0.03% | 0.05% | 0 | 0 | 0 | 0 | 0 | 0 | 0.01% | 0.03% |
| ES19 | 0.62% | 1.39% | 0 | 0 | 6.75% | 2.09% | 11.95% | 12.85% | 0.01% | 0 |
| ES20 | 88.34% | 98.55% | 0 | 0 | 0 | 0 | 0 | 0 | 0.01% | 0 |
| ES21 | 0.01% | 0.01% | 0 | 0 | 0 | 0 | 56.44% | 37.37% | 0 | 0 |
| ES22 | 1.08% | 1.31% | 0 | 0 | 6.74% | 8.94% | 0 | 0 | 0 | 0 |
| ES23 | 0.07% | 0.34% | 0 | 0 | 0.01% | 0 | 0 | 0 | 0.16% | 0.23% |
| ES24 | 0.07% | 0.15% | 0 | 0 | 0.60% | 0.28% | 0.01% | 0 | 87.24% | 81.12% |
| ES25 | 0 | 0 | 0 | 0 | 1.18% | 0.41% | 2.41% | 1.51% | 0 | 0 |
| ES26 | 41.52% | 87.96% | 0 | 0 | 5.60% | 0.64% | 0 | 0 | 0 | 0 |
| ES27 | 64.41% | 97.56% | 0 | 0 | 0.43% | 0.02% | 2.93% | 0.11% | 0 | 0 |
| ES28 | 0.02% | 0.04% | 0 | 0 | 0.01% | 0 | 0.01% | 0.04% | 0.02% | 0.01% |
| ES29 | 73.98% | 98.00% | 0.01% | 0 | 0 | 0 | 0 | 0 | 0.05% | 0.01% |
| ES30 | 72.00% | 87.13% | 0 | 0 | 0 | 0 | 0 | 0 | 0.18% | 0.14% |
| ES31 | 90.32% | 96.74% | 0 | 0 | 0 | 0 | 0.01% | 0.01% | 0 | 0 |
| ES32 | 0.02% | 0.01% | 0 | 0 | 0.02% | 0 | 33.62% | 57.14% | 0.07% | 0.04% |
| ES33 | 49.36% | 78.78% | 0 | 0 | 0 | 0 | 0.07% | 0 | 0.03% | 0.06% |
| ES34 | 0.05% | 0.15% | 0 | 0 | 0 | 0 | 0 | 0 | 0 | 0.03% |
| ES35 | 1.84% | 7.29% | 0 | 0 | 0.66% | 0.40% | 0.17% | 0.10% | 22.94% | 21.79% |

**Table 3** (*continued*)

| | *Staphylococcus* | | *Achromobacter* | | *Haemophilus* | | *Stenotrophomonas* | | *Pseudomonas* | |
|---|---|---|---|---|---|---|---|---|---|---|
| | Non EnzD | EnzD | Non EnzD | EnzD | Non EnzD | EnzD | Non EnzD | EnzD | Non EnzD | EnzD |
| ES36 | 0.30% | 0.16% | 0 | 0 | 0.32% | 0.02% | 0.01% | 0.02% | 28.07% | 96.42% |
| ES37 | 0 | 0.02% | 0 | 0 | 1.90% | 1.22% | 0 | 0 | 0.18% | 0.31% |
| ES38 | 0.16% | 0.53% | 0.08% | 0.10% | 0.05% | 0.10% | 0 | 0 | 0.01% | 0.03% |
| ES39 | 0.01% | 0 | 0 | 0 | 0 | 0 | 0 | 0 | 0 | 0.01% |
| | 15.3 ($-$14.6, 64.8) | | $-$ | | $-$3.9 ($-$7.9, 2.2) | | $-$2.8 ($-$19.1, 23.5) | | 1.1 ($-$42.0, 68.4) | |

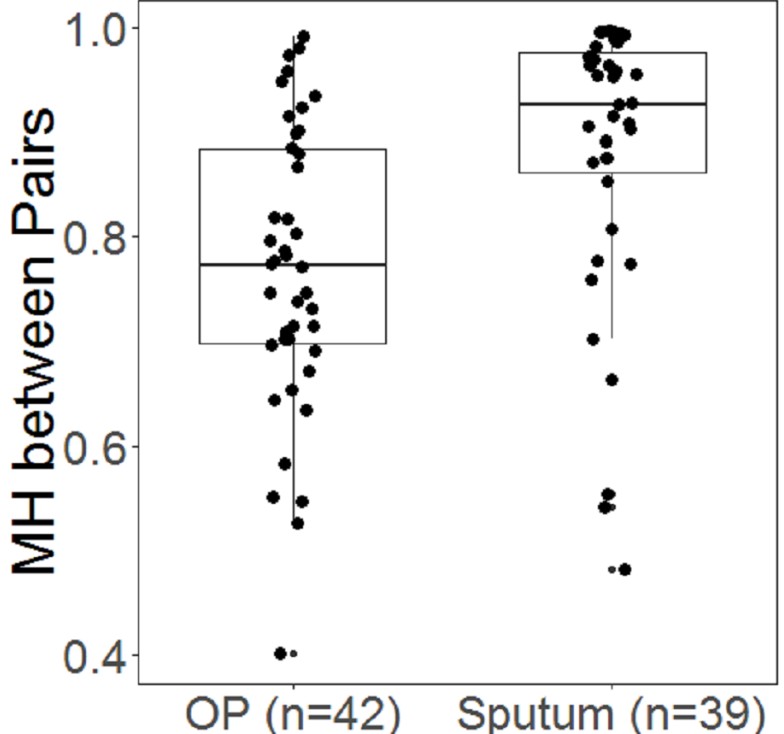

**Figure 2** **Boxplots showing distribution of Morisita-Horn (MH) similarity metric for paired samples in OP and sputum.** Wilcoxon rank sum tests indicate the distributions between the two sample types were statistically different ($p < 0.01$). The MH metric ranges from 0 to 1 with a value of 1 indicating identical communities, and 0 indicating no overlap between communities. The line indicates the median value and the box ranges from the 25th and 75th percentiles, the whiskers extend to 1.5 times the interquartile range.

demonstrated large changes in MH with EnzD, but with no consistent direction of change as was seen in the OP samples (Fig. 3B). Sputum appears to be more heterogeneous compared to OP samples. Sputum was primarily influenced by enhanced extraction of *Staphylococcus, Veillonella,* and *Streptococcus* (Fig. 3B). This result is consistent with Table 2 where *Staphylococcus* and *Veillonella* are shown to have a larger change than *Streptococcus*.

## Associations between community composition and MH

We found no striking evidence of a single phylum or genus whose relative abundance was associated with MH (Fig. S5). Combination of Gram-positive bacteria was calculated using

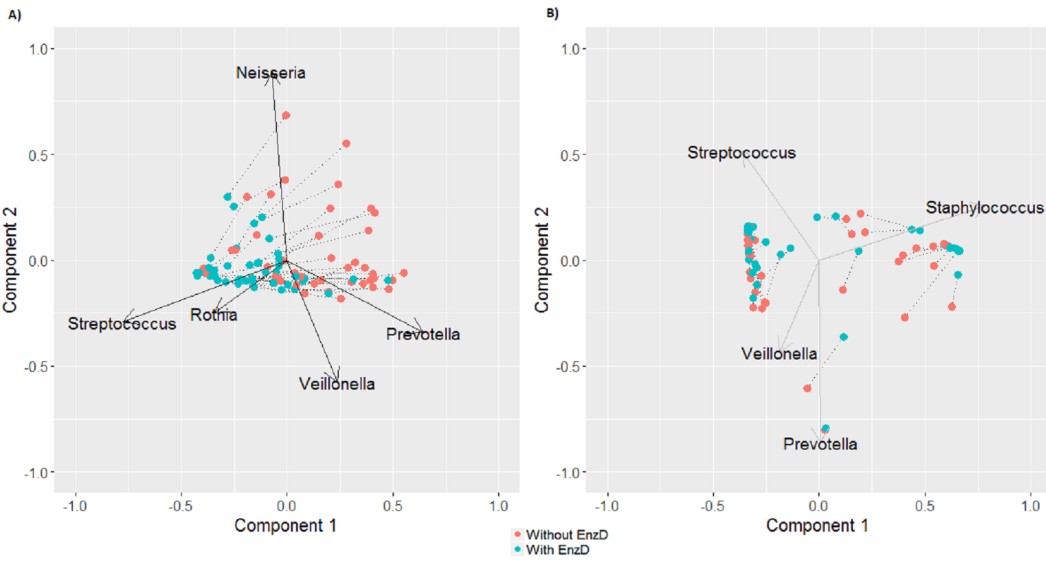

**Figure 3** **Ordination biplot, using 1-MH beta diversity values for OP (A) and sputum (B).** Values closer together are more similar. Paired samples are connected with a line, vectors for the genera with at least 1% RA and $p < 0.005$ (by permutation test) for $R^2$ of the loading.

the summation of (Actinobacteria + Firmicutes (-*Veillonella*)). For the OP samples with large changes in their communities with EnzD, the relative abundance of Gram-positive organisms was lower compared to those samples with limited changes in their communities (Fig. 4). For all the OP samples, the relative abundance of Gram-positive organisms was higher with EnzD. There was an association between change in relative abundance of Gram-positive organisms with EnzD and MH values for the pairs (Spearman's correlation, $r = -0.60$; $p < 0.01$; Fig. 4B).

## DISCUSSION

In this work, we demonstrate that EnzD of airway samples changes the bacterial community detected by sequencing primarily due to increased detection of organisms with a Gram-positive cell wall structure. From the biplots (Fig. 3A) we can see that for OP samples there is a clear shift in microbial composition between those without EnzD and those with EnzD. With EnzD the OP samples are highly clustered around *Gemella*, *Rothia*, and *Streptococcus* whereas those without EnzD are dispersed around *Neisseria, Prevotella* and *Veillonella*. These observations are consistent with cell wall structure; we would predict that given their rigid cell wall Gram-positive bacteria would drive changes in community composition for those samples with EnzD. *Streptococcus*, *Gemella*, and *Rothia* explain a large amount of the variability in the OP samples. *Streptococcus* and *Staphylococcus* are Gram-positive taxa explaining a large amount of the variability in the sputum samples. Among the Gram-negative taxa explaining a large amount of variability in OP and sputum samples are *Neisseria*, *Veillonella*, and *Prevotella* due to decreasing relative abundance with EnzD. *Granulicatella*, a member of the Firmicutes had decreased RA with EnzD, which was not consistent with other Gram positive taxa. *Granulicatella* is reported to have
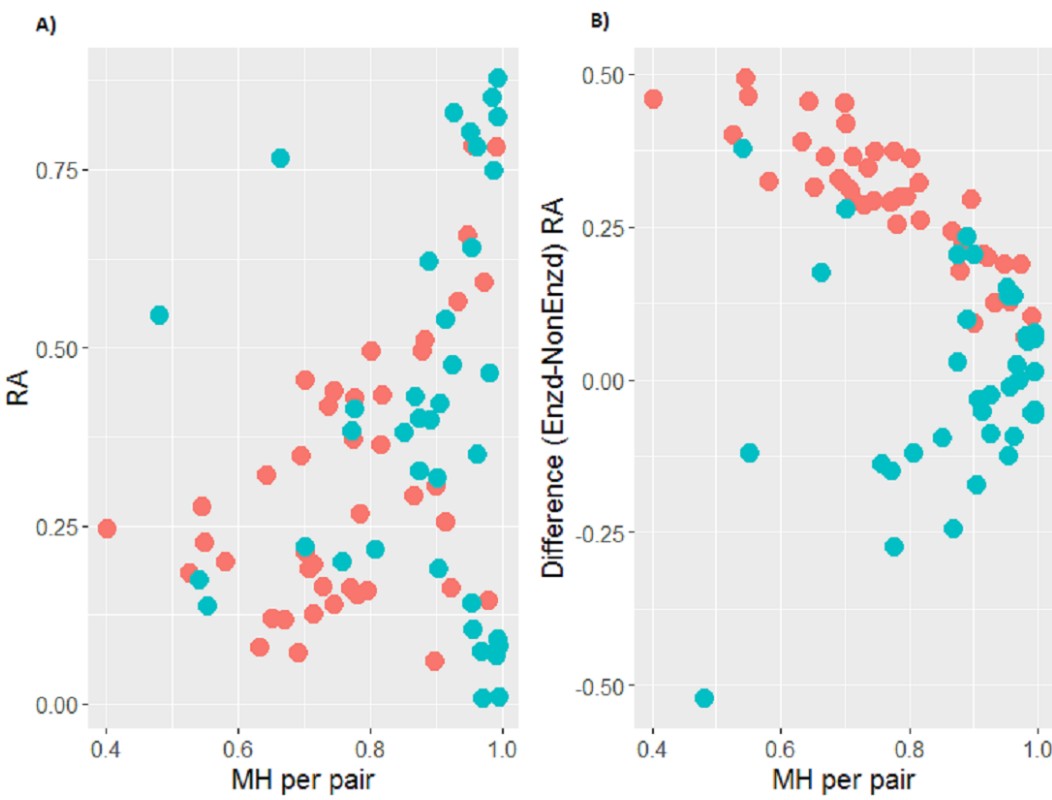

**Figure 4** **(A) Scatterplot of relative abundance of Gram-positive taxa in non-EnzD samples versus Morisita-Horn (MH) similarity metric of paired samples. MH metric ranges from 0 to 1 with a value of 1 indicating identical communities, and 0 indicating no overlap between communities. OP samples are shown in red and sputum in blue. (B) Scatterplot of the difference in relative abundance of Gram-positive taxa versus MH of paired samples. OP samples are shown in red and sputum in blue.**

pleomorphic and variable Gram stain characteristics, which may explain this observation (*Bottone et al., 1995*; *Christensen & Facklam, 2001*).

Difference in OP and sputum samples can be due to differences in the microbial communities which may be related to multiple factors including anatomic location, inter individual variability and disease severity. The samples used in this study were collected for clinical evaluation of airway infection, although separate evaluation for sample adequacy was not performed, these samples are reflective of what is being used to make clinical decisions. The patients that were not capable of expectorating sputum tend to be younger with presumably less lung disease (i.e., fewer pathogens). There are also differences between upper and lower airways (*Zemanick et al., 2015*) that could also impact the community composition observed.

*Willner et al. (2012)* demonstrated that genera detection varied significantly between five commonly used extraction methods with *Staphylococcus* being detected with varying efficiency. Support for these findings showed the largest difference in microbial proportions between two DNA extraction methods (Norgen and Qiagen) was found in *Staphylococcus* (*Pérez-Losada et al., 2016*). Consistent with previous findings, EnzD revealed a higher RA of *Staphylococcus* in OP samples. Further, when evaluated in both OP swab and

sputum samples from pediatric subjects with CF, variations in detection of *Staphylococcus* were observed (*Johnson et al., 2016*). Total RA for *Staphylococcus* in sputum samples was higher overall, but had less of a change before and after EnzD (*Johnson et al., 2016*). *Zhao et al. (2012)* looked at sputum and saw an increase in *Staphylococcus*, which we corroborate. We also found a higher RA of *Streptococcus* in OP samples. *Johnson et al. (2016)* found that for OP samples DNA concentration consistently increased with EnzD. This indicates that in addition to observing changes in RA, EnzD probably better represents the absolute amounts of genera present in OP samples.

### Limitations

There are certain limitations to our study. First, only two airway sample types collected from subjects with CF were evaluated, thus limiting more general inferences regarding the effects of EnzD in other samples types. However, this study does include OP samples, which have not previously been evaluated and represents an important sample type in early CF. The patient's clinical status at the time of sample collection is unknown, this hampers our ability to determine whether these samples are reflective of both clinical stability or during a pulmonary exacerbation. However, because paired samples were used to assess the effect of EnzD, the difference for each pair is not confounded by clinical status. Because only a single replicate with and without EnzD was evaluated, we are unable to compare effects due to EnzD versus technical variability. However, the observation of the consistent differences for the majority of the sample pairs provides some evidence that the change is due to EnzD rather than technical variability. In our study we considered beta-diversity values of 0.8 or greater to be within the limits of change due to biological variability.

The utility of EnzD is difficult to predict due to the strong influence of the community composition in any given sample on the results. There is a compelling argument to utilize EnzD in studies that rely on OP swabs as previous studies have shown increased sensitivity for detection of Staphylococcus. While the argument is less compelling for sputum, EnzD should be applied consistently. Other sample types would require evaluation to determine how much impact is observed. However, increasing the RA of any group makes it harder to identify minor components in the community. EnzD could negatively impact the ability to identify low abundance Gram-negative organisms as seen through the decrease in Gram-negative genera shown in Table 2. This includes many important pathogens in CF (e.g., *Pseudomonas*) and should be considered carefully.

## CONCLUSIONS

In summary, we found that the use of EnzD on airway samples prior to DNA extraction and sequencing alters microbiome community composition results, likely due to improved detection of gram-positive bacterial taxa (e.g., *Streptococcus* and *Staphylococcus*). The use of EnzD appears particularly important for analysis of OP swab samples. EnzD use with sputum samples may be less critical and has the potential to decrease sensitivity for low abundance taxa. Our findings highlight the need for a consistent approach to airway sample processing and analysis, and suggest that EnzD should be applied routinely in studies using OP swabs and studies requiring sensitive detection of *Staphylococcus* and *Streptococcus*.

### Funding

The authors received no funding for this work.

### Competing Interests

The authors declare there are no competing interests.

### Author Contributions

- Kayla M. Williamson analyzed the data, wrote the paper, prepared figures and/or tables, reviewed drafts of the paper.
- Brandie D. Wagner wrote the paper, prepared figures and/or tables, reviewed drafts of the paper.
- Charles E. Robertson analyzed the data, contributed reagents/materials/analysis tools, reviewed drafts of the paper.
- Emily J. Johnson conceived and designed the experiments, performed the experiments, reviewed drafts of the paper.
- Edith T. Zemanick conceived and designed the experiments, contributed reagents/materials/analysis tools, reviewed drafts of the paper.
- J. Kirk Harris conceived and designed the experiments, performed the experiments, contributed reagents/materials/analysis tools, wrote the paper, prepared figures and/or tables, reviewed drafts of the paper.

### Human Ethics

The following information was supplied relating to ethical approvals (i.e., approving body and any reference numbers):

The Colorado Multiple Institutional Review Board approved the study (COMIRB 07-0835). Written informed consent was obtained from patients or guardians. Written informed assent was obtained for children 10–17 years of age.

### Data Availability

Sorted paired end sequence data were deposited in the NCBI Short Read Archive under accession number SRP043334.

### Supplemental Information

Supplemental information for this article can be found online at http://dx.doi.org/10.7717/peerj.3362#supplemental-information.

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
