# Peer review of "Impact of enzymatic digestion on bacterial community composition in CF airway samples"

_PeerJ, doi:10.7717/peerj.3362_

## Round 0.1 · original submission · Major Revisions

As you can see, your submission was examined by 3 reviewers who essentially concluded that the study was of interest but that the paper required a series of important corrections. They also concluded that a revision was doable. So, please, take the time to examine in details the suggestions made and the criticisms raised and provide a detailed rebuttal in which you indicate exactly how you have dealt with them.

Reviewer 1 ·

Basic reporting

No comment -- see general comments

Experimental design

The authors do not mention correcting p values for multiple comparisons following Wilcoxon Signed rank testing of the relative abundances for genera and phyla. This should be performed and corrected p values reported.

Validity of the findings

No comment

Additional comments

I find myself distracted by the large range in patient age and wondering whether the initial alpha diversity of the samples [e.g. younger patients have higher diversity in sputum (Cox et al. 2010)] contribute to the overall effect of the enzD. In Fig. 3B, a subset of patients clustered tightly together – the authors briefly mention this group as primarily represented by Veillonella, but not whether these are adult patients with low initial diversity or whether other Gram- taxa were dominant.

The authors mention that there are three distinct clusters in the sputum samples. I think it would be worthwhile to report the age distribution and average alpha-diversity of the patients within each cluster and to validate these clusters statistically (or remove mention of the clusters).

Shannon diversity was reduced with enzD in sputum samples (and, to some extent, the OP samples). I agree that this is likely due to increased relative abundance of Gram+ and a less even community, but it would be helpful to untangle the richness and evenness and calculate these metrics separately (richness and Pielou’s Evenness). I’d hypothesize that richness would increase in enzD-treated samples while evenness would decrease.

“Standard homogenization” procedures were used to process sputum samples. What does this refer to? Did they use sputolysin to homogenize the samples?

The legends for Figures 2 and 4 could use more detail.

Figure 2 would benefit from an arrow along the y-axis to indicate that values approaching 0 are less similar and values approaching 1 are more similar. The MH index is opposite some of the other commonly used beta-diversity indices used in microbiome studies where 0=identical and 1=no overlap. The authors should also include the p value on this figure/within the legend.

The authors discuss in the Materials and Methods that these samples were also sent for standard culture. Did enzD increase molecular detection of Gram positive taxa in specimens that also cultured G+ bacteria?

Mechanical lysis (bead-beating) is another commonly used modification to DNA extraction protocols and can help break apart fungal cell walls. There is no mention of alternative lysis methods throughout the manuscript, and I wonder whether mechanical lysis in the presence or absence of enzD can also increase abundance of gram positive taxa within a sample. The authors should discuss mechanical lysis methods in the context of their results in the discussion.

Line 79: 16S rRNA gene Amplicon Library Construction.

Overall, the conclusions are well supported by the results presented here. The consistent shift in OP microbial community composition and increase in the relative abundance of Streptococcus representation is beautifully demonstrated in Figure 3A, though Fig. 3B is more difficult to draw a meaningful conclusion from. Despite this, I think the authors do a nice job of interpreting these results.


References cited in the comments above:
Cox, Michael J., Martin Allgaier, Byron Taylor, Marshall S. Baek, Yvonne J. Huang, Rebecca A. Daly, Ulas Karaoz, et al. 2010. “Airway Microbiota and Pathogen Abundance in Age-Stratified Cystic Fibrosis Patients.” Edited by Adam J. Ratner. PLoS ONE 5 (6). Public Library of Science: e11044. doi:10.1371/journal.pone.0011044.
Knudsen, Berith E., Lasse Bergmark, Patrick Munk, Oksana Lukjancenko, Anders Priemé, Frank M. Aarestrup, and Sünje J. Pamp. 2016. “Impact of Sample Type and DNA Isolation Procedure on Genomic Inference of Microbiome Composition.” mSystems 1 (5).
Wagner Mackenzie, Brett, David W Waite, and Michael W Taylor. 2015. “Evaluating Variation in Human Gut Microbiota Profiles due to DNA Extraction Method and Inter-Subject Differences.” Frontiers in Microbiology 6. Frontiers Media SA: 130. doi:10.3389/fmicb.2015.00130.
Yuan, Sanqing, Dora B Cohen, Jacques Ravel, Zaid Abdo, and Larry J Forney. 2012. “Evaluation of Methods for the Extraction and Purification of DNA from the Human Microbiome.” PloS One 7 (3): e33865. doi:10.1371/journal.pone.0033865.
Zhao, Jiangchao, Lisa A. Carmody, Linda M. Kalikin, Jun Li, Joseph F. Petrosino, Patrick D. Schloss, Vincent B. Young, et al. 2012. “Impact of Enhanced Staphylococcus DNA Extraction on Microbial Community Measures in Cystic Fibrosis Sputum.” Edited by John R. Battista. PLoS ONE 7 (3). Public Library of Science: e33127. doi:10.1371/journal.pone.0033127.

Reviewer 2 ·

Basic reporting

Adequate, except does not do a good job citing prior work, explaining EnzD, or even in discussing why it would be important to look at an OP sample in a CF patient.

Experimental design

Lacks validation of the role of EnzD. Maybe this is due to lack of literature cited, but would be important to know if the enhancement of gram positive organisms after EnzD digestion reflects true relative abundances or something else.

Validity of the findings

The S. aureus vs other staphylococci question is important to distinguish if possible.

Additional comments

1. What this study is missing is some form of validation. For example, it is noted that S. aureus detection is enhanced by the use of EnzD. Is it really that we are seeing it's true relative abundance because of the use of EnzD, or is it because other bacteria relatively speaking are of lower abundance after EnzD digestion? Would be nice for the authors to perhaps spike some samples with known concentrations of some mock communities to decipher whether EnzD digestion gets us closer to the true relative abundances of these bacteria, or whether it simply enhances the relative abundance of some singular bacteria while not truly reflecting their relative abundances. Perhaps this data already exists, and if it does, the authors should cite those articles and describe in the introduction.

2. In the introduction, the authors don't really state what EnzD does? Is it digesting the cell wall carbohydrates, etc...?

3. The methods section does not cover how the EnzD digestion was performed.

4. Why the choice of Morisita Horn beta diversity? This is not common, so the authors should provide an explanation. It would definitely be helpful to know how this choice of distance measures may affect the PCOA analysis.

5. I don't fully understand the paragraph from lines 140-151. It states that Prevotella, Fusobacterium, etc... may not be clinically meaningful, which is quite a vague statement. Later, it points to an increase in the RA for Staphylococcus with the suggestion that the Staphylococcus is clinically meaningful. Based on the species of Staphylococcus present, I'm not sure it would be any more clinically meaningful than the other examples. Aren't we really asking is this S. aureus or other staphylococcal species? I'm not sure if the V1-V2 region is sufficient to know this answer, but is hard to read about an increase in the RA of Staphylococcus without wondering whether this specifically refers to a pathogen like S. aureus or mostly nonpathogens such as S. epidermidis.

6. The Shannon Index I don't believe is a good measure of diversity in these situations. It does not penalize for the relatively low abundant species present, and thus, you will have great shifts in diversity when using the EnzD and some species become much more abundant. Probably better to use several different diversity measures to show they all show the same trend, or to switch to a different measure that penalizes for a higher number of lower abundance species.

7. With regards to the sputum samples, was there any quality control to ensure the sputum samples were good specimens? Most of the bacteria identified in this study are oral microbes. The authors should probably explain why so many oral microbes are identified in sputum, and whether this is a common feature of the CF lung compared to healthy individuals.

8. Any explanation for the paradoxical values observed with Granulicatella in response to EnzD?

Reviewer 3 ·

Basic reporting

The manuscript "Impact of enzymatic digestion on bacterial community
composition in CF airway samples" by Williamson et al. describes the impact of prior preparation of airway samples (n=81) including oropharyngeal swabs and sputa of 63 patients by enzymatic digestion and its impact on relative abundance of bacterial genera and species. The authors show that the alpha-diversity decreased in oropharyngeal swabs and sputum samples while larger shifts in community composition were observed for oropharyngeal swabs compared to sputa. Using prior enzymatic digestion the authors detected a higher relative abundance of grampositive species such as Gemella, Streptococcus and Rothia in oropharyngeal samples, while from sputa higher relative abundance was only detected for S. aureus, whereas other species such as Granulicatella, Veillonella, Gemella and Prevotella were less abundant.
This is an interesting study. However, there are major and minor points of criticism.

Experimental design

well done

Validity of the findings

Major points of criticism:
1. The authors show that with their protocol of enzymatic digestion there were shifts in community composition. However, they did not report changes in the relative abundance of important CF-pathogens such as Pseudomonas aeruginosa, Stenotrophomonas maltophilia, Burkholderia cepacia complex. The authors should comment on this.
2. Since S. aureus was the only species with higher relative abundance in sputa as the authors reported the conclusion that enzymatic digestion should be especially used for oropharyngeal swabs in order to optimize detection of S. aureus is not justified.
3. The authors report about 7 samples that were negative for "standard" CF pathogens. The authors should define what "standard" CF pathogens are. Furthermore, they should show the results from sequencing for these specimens because these data are especially interesting. How is the clinical course in these patients? Are these specimens from routine visits? Or from exacerbations?
4. Line 143: The authors report about less abundance of some bacterial species. Here, they also report about less abundance of Haemophilus, which is especially in young patients considered as a real CF pathogen. The authors should stress this finding.
Minor points of criticism:
1. Once "Staphylococcus aureus" has been introduced, the authors should only write "S. aureus". This also should be changed for all bacteria.
2. References: There are unusual terms in the reference list, which do not belong there, e.g.:
LiPuma, J. J. 2010. “The Changing Microbial Epidemiology in Cystic Fibrosis.” Clinical
279 Microbiology Reviews 23(2):299–323. Retrieved August 23, 2016
280 (http://cmr.asm.org/cgi/doi/10.1128/CMR.00068-09).

---

## Round 0.2 · Minor Revisions

As you can see, your revised manuscript was examined by two of the original reviewers with one of them suggesting additional corrections. Please, follow these suggestions and submit a new version.

Reviewer 1 ·

Basic reporting

The authors have made several changes to this manuscript, including clarification of the methods and expanding the background. They also calculated additional alpha diversity metrics to supplement their findings that were significant using Shannon’s diversity metric. I have minor comments, mostly clarification of statistical tests, outlined below.

Experimental design

no comment

Validity of the findings

Are the panels swapped in Fig. 3? The text states that 3A are OP and 3B are sputum, but the legend has it the opposite. This figure is the most convincing, in my opinion, so if the legend is correct, then the conclusions need to be revisited.

Additional comments

The authors state in their response that they corrected p values for multiple comparisons using the benjamini-hochberg method. However, their tables and figures (Table 1 and 2 and Figure 1) do not reflect the new p values. They should, at the very least, report the corrected and uncorrected p value.

Throughout the manuscript, the statistical results within parentheses should be formatted to describe the test and metric in addition to the p value. E.g. Page 10, line 4, instead of (p<0.01) should be (Shannon Diversity, nonparametric t test, p<0.01).

The authors don’t describe which test was used for the alpha diversity comparisons in the Statistical Analysis section of the methods.

Figure 3 legend – please state the statistical test (spearman correlation) associated with the reported p value.

Reviewer 3 ·

Basic reporting

no comment

Experimental design

no comment

Validity of the findings

no comment

Additional comments

The revised version of the manuscript improved by the changes introduced due to the reviewer's comments.

---

## Round 0.3 · accepted · Accept

Thank you also for taking into account the comments raised on your resubmission.